# FlashEVA: Accelerating LLM inference via Efficient Attention

## Abstract

Transformer models have revolutionized natural language processing, achieving state-of-the-art performance and demonstrating remarkable scalability. However, their memory demands, particularly due to maintaining full context in memory, pose significant challenges for inference. In this paper, we present FlashEVA, an efficient implementation of EVA (Efficient Attention via Control Variates), and demonstrate how to finetune transformers to adapt to FlashEVA attention. Our method enables fine-tuning of Transformer models with as few as 1.5B tokens while preserving effectiveness across various downstream tasks. Notably, FlashEVA achieves up to 6.7x higher throughput and 5x lower peak GPU memory usage during inference compared to standard Transformer implementations. Despite these improvements, we observe limitations in retrieval-focused tasks. Our implementation offers control over the trade-off between throughput and accuracy through adjustable hyperparameters, providing flexibility for diverse use cases. This work represents a significant step towards more efficient and adaptable Transformer-based models for inference.

## 1 Introduction

Transformer models have become ubiquitous in the field of natural language processing, achieving state-of-the-art performance across a wide range of tasks (Dosovitskiy et al., 2021; Wang et al., 2019; Radford et al., 2019; Dong et al., 2018). Their success can be attributed to their ability to scale effectively and the possibility of parallel training, which has led to significant improvements in model capabilities (Kaplan et al., 2020; Gadre et al., 2024). However, these advancements come at a cost: Transformers have high memory requirements during inference, particularly due to the need for an ever-increasing cache to keep the full history in context (Pope et al., 2022).

This memory constraint poses a significant challenge, especially as new use cases emerge that demand higher throughput. Such applications include inference on long contexts (e.g. Q&A over long documents or large codebases) (Rozière et al., 2024; Li et al., 2023; Guo et al., 2022; Shaham et al., 2022), agentic workflows that require exploring multiple trajectories in parallel (Yang et al., 2024a; Yao et al., 2023b), as well as workflows combining search algorithms with LLMs (Chen et al., 2023; Yao et al., 2023a; Wang & Zhou, 2024)

As these demands grow, it becomes increasingly important to address the limitations of Transformer models. A crucial consideration is the desire to leverage the existing pre-trained Transformer models, avoiding the need for resource-intensive retraining from scratch. There are different approaches that aim to address these issues:

- **Distributed KV cache storage**, as shown by Liu et al. (2023a), can be achieved without communication overhead, due to the fact that the attention operation can be computed in a blockwise fashion (Dao et al., 2022). This allows the computation of different blocks to be done on separate devices and can in principle support an infinite context length provided enough GPUs are available.

- **KV cache compression** techniques aim to compress the existing KV cache Zhao et al. (2024); Yue et al. (2024), or discard uninformative tokens from the cache (Jiang et al., 2023; Han et al., 2024) to minimize the memory cost without sacrificing model performance.

- **State-Space Models (SSMs)** are an alternative class of models that have shown to be competitive with transformers on small to medium scale (Gu & Dao; Yang et al., 2023; Qin et al., 2024b; Peng et al., 2023). Recent works Wang et al. (2024); Bick et al. (2024) have shown that transformers can be efficiently distilled into Mamba, keeping most of the downstream performance.

- **Linearized attention** approaches aim to replace the Softmax attention in the transformer with linearized variants, which promises to reduce the computational and memory requirements of the attention (Zheng et al., 2023; Qin et al., 2022; Peng et al., 2021). Previous works attempting to adapt pretrained transformers into linearized transformers have not scaled or maintained model performance on downstream tasks (Chen et al., 2024; Mao, 2022; Kasai et al., 2021a).

In this paper, we propose to revisit and enhance the Efficient Attention via Control Variates (EVA) method (Zheng et al., 2023) to address these challenges. We present an efficient implementation of EVA attention using custom CUDA and Triton kernels, which allows us to maintain the benefits of Transformer models while reducing their memory and computational footprint.

Our approach demonstrates that Transformers can be fine-tuned with as little as 1.5 billion tokens, recovering most of the original performance on downstream tasks. Nevertheless, the performance still suffers on retrieval focused tasks, similar to previous approaches with linearized attention and state space models (Waleffe et al., 2024).

While FlashEVA attention still requires to keep a cache of (compressed) past context, this overhead is less than the memory overhead for computing the prefix during model inference, and is thus does not represent a limitation for the inference. Notably, compared to vanilla transformers, our method achieves significant improvements in inference scenarios of particular interest, such as generating content with long prompts, handling extended generations, and maximizing parallel throughput. We report up to $6.7x$ inference throughput increase on long sequence generation, and up to $5x$ reduction in peak GPU memory usage on long sequence generation. Additionally, EVA attention allows for trading off memory and model performance in a principled manner, through two hyperparameters, allowing up to $50\%$ lower memory usage with $0.5\%$ accuracy impact on downstream task performance.

The remainder of this paper is structured as follows: In Section 2, we review the theoretical foundations of EVA attention and introduce our efficient implementation, which we call FlashEVA. Section 3 details our experimental setup, including model architecture, training procedures, and evaluation metrics. Section 4 presents our results, comparing FlashEVA's performance to existing methods across various tasks and inference scenarios. We also provide an in-depth analysis of the trade-offs between speed, accuracy, and memory usage. Finally, Section 5 concludes the paper, summarizing our findings and discussing potential avenues for future research in efficient attention mechanisms for large language models. An extended review of related works, as well as ablation studies are in the Appendix.

## 2 BACKGROUND

This section provides a comprehensive review of key attention mechanisms, establishing the foundation for our work. We begin by revisiting the Softmax attention mechanism and elucidating its relationship to Random Feature Attention. Subsequently, we examine the Efficient Vision Attention (EVA) framework, which reinterprets Randomized Attention through the lens of control variates. Finally, we highlight the practical implementation challenges associated with EVA attention and demonstrate how these limitations can be addressed by reformulating the approach as Softmax attention over a modified set of keys and values.

### 2.1 SOFTMAX ATTENTION

Softmax attention is a fundamental component of transformer architectures. Let $\boldsymbol{Q} \in \mathbb{R}^{N \times D}$, $\boldsymbol{K} \in \mathbb{R}^{M \times D}$, and $\boldsymbol{V} \in \mathbb{R}^{M \times D}$ denote the query, key, and value matrices, respectively, where $N$ is the number of queries, $M$ is the number of keys and values, and $D$ is the dimensionality of the

embedding space.[1] The Softmax attention mechanism for a single query vector $\boldsymbol{q}_n$ (the $n$-th row of $\boldsymbol{Q}$) can be expressed as:

$$\text{SoftmaxAttn}(\boldsymbol{q}_n, \boldsymbol{K}, \boldsymbol{V}) := \sum_{m=1}^{M} \frac{\exp(\boldsymbol{q}_n^\top \boldsymbol{k}_m)}{\sum_{m'=1}^{M} \exp(\boldsymbol{q}_n^\top \boldsymbol{k}_{m'})} \boldsymbol{v}_m^\top = \sum_{m=1}^{M} \frac{\text{sim}(\boldsymbol{q}_n, \boldsymbol{k}_m)}{\sum_{m'=1}^{M} \text{sim}(\boldsymbol{q}_n, \boldsymbol{k}_{m'})} \boldsymbol{v}_m^\top \quad (1)$$

where $\boldsymbol{k}_m$ and $\boldsymbol{v}_m$ are the $m$-th rows of $\boldsymbol{K}$ and $\boldsymbol{V}$, respectively, and $\text{sim}(\cdot, \cdot)$ denotes the similarity function, typically implemented as a dot product. In the case of causal attention, the summation is restricted to $m \leq n$, ensuring that each token attends only to previous tokens and itself. It is important to note that computing the attention output for the latest query necessitates a summation over all previous timesteps, which is the reason why it is memory intensive to run inference with Softmax attention.

## 2.2 RANDOM FEATURE ATTENTION

Random Feature Attention, introduced by Peng et al. (2021) and Choromanski et al. (2022), leverages random feature methods (Rahimi & Recht, 2007) to linearize the exponential kernel in attention mechanisms. This approach approximates the exponential kernel using the following expectation:

$$\exp(\mathbf{x}^\top \mathbf{y}) = \mathbb{E}_{\omega \sim \mathcal{N}(0,\mathbf{I})} \left[ \xi(\mathbf{x}, \omega)^\top \xi(\mathbf{y}, \omega) \right] \approx \frac{1}{S} \sum_{s=1}^{S} \xi(\mathbf{x}, \omega_s)^\top \xi(\mathbf{y}, \omega_s) \quad (2)$$

where $\xi(\cdot, \cdot) : \mathbb{R}^D \times \mathbb{R}^D \to \mathbb{R}^l$ is a randomized mapping that projects the input into a random feature space. While the exact formulation of this mapping differs slightly between Peng et al. (2021) and Choromanski et al. (2022), we adopt the latter's approach, defining $\xi(\mathbf{x}, \omega) = \exp\left(\omega^\top \mathbf{x} - \frac{1}{2}\|\mathbf{x}\|^2\right)$. Employing this random feature mapping, we can reformulate the standard attention mechanism as Random Feature Attention:

$$\sum_{m=1}^{M} \frac{\exp(\mathbf{q}_n^\top \mathbf{k}_m)}{\sum_{m'=1}^{M} \exp(\mathbf{q}_n^\top \mathbf{k}_{m'})} \mathbf{v}_m^\top \approx \frac{\sum_{s=1}^{S} \xi(\mathbf{q}_n, \omega_s)^\top \sum_{m=1}^{M} \xi(\mathbf{k}_m, \omega_s)\mathbf{v}_m^\top}{\sum_{s=1}^{S} \xi(\mathbf{q}_n, \omega_s)^\top \sum_{m'=1}^{M} \xi(\mathbf{k}_{m'}, \omega_s)} := \text{RFA}(\mathbf{q}_n, \mathbf{K}, \mathbf{V}) \quad (3)$$

### 2.2.1 RANDOMIZED ATTENTION

Furthermore, Zheng et al. (2022) demonstrate that the Softmax attention can be expressed as the following expectation:

$$\text{SoftmaxAttn}(\mathbf{q}_n, \mathbf{K}, \mathbf{V}) = \mathbb{E}_{p_n(\omega)}[f_n(\omega)] = \mathbb{E}_{p_n(\omega)} \left[ \frac{\xi(\mathbf{q}_n, \omega)^T \sum_{m=1}^{M} \xi(\mathbf{k}_m, \omega)\mathbf{v}_m^T}{\xi(\mathbf{q}_n, \omega)^T \sum_{m'=1}^{M} \xi(\mathbf{k}_{m'}, \omega)} \right] \quad (4)$$

where $p_n(\omega) = \sum_{m=1}^{M} \pi_m \mathcal{N}(\omega; \mathbf{q}_n + \mathbf{k}_m, \mathbf{I})$ represents the proposal distribution and $\pi_m = \frac{\exp(\mathbf{q}_n^T \mathbf{k}_m)}{\sum_{m'=1}^{M} \exp(\mathbf{q}_n^T \mathbf{k}_{m'})}$ denotes the component weight. The RFA attention can thus be interpreted as performing self-normalized importance sampling of Equation 4, utilizing $q(\omega) = \mathcal{N}(\omega; 0, \mathbf{I})$ as the proposal distribution. Consequently, the equivalent RFA formulation can be expressed as:

$$\text{RFA}(\mathbf{q}_n, \mathbf{K}, \mathbf{V}) = \frac{\mathbb{E}_{q(\omega)} \left[ \frac{p_n(\omega)}{q(\omega)} f_n(\omega) \right]}{\mathbb{E}_{q(\omega)} \left[ \frac{p_n(\omega)}{q(\omega)} \right]} \approx \frac{\sum_{s=1}^{S} \frac{p_n(\omega_s)}{q(\omega_s)} f_n(\omega_s)}{\sum_{s=1}^{S} \frac{p_n(\omega_s)}{q(\omega_s)}} \quad (5)$$

## 2.3 EVA ATTENTION

Zheng et al. (2023) demonstrate that RFA attention in the SNIS formulation (Equation 5) can be rewritten as a sum of control variate estimates. By altering the order of summation, we obtain:

$$g(\omega) = \sum_{s=1}^{S} \frac{p_n(\omega_s)}{q(\omega_s)} f_n(\omega_s) = \sum_{s=1}^{S} \frac{1}{S} \sum_{m=1}^{M} \frac{\mathcal{N}(\omega_s; 0, I)}{Z q(\omega_s)} \xi(q_n, \omega_s)\xi(k_m, \omega_s)v_m = \sum_{m=1}^{M} g_m(\omega) \quad (6)$$

---

[1]For simplicity, we omit the batch and head dimensions in our formulations.

$$h(\omega) = \sum_{s=1}^{S} \frac{p_n(\omega_s)}{q(\omega_s)} = \sum_{s=1}^{S} \frac{1}{S} \sum_{m=1}^{M} \frac{\mathcal{N}(\omega_s; 0, I)}{Z q(\omega_s)} \xi(q_n, \omega_s) \xi(k_m, \omega_s) = \sum_{m=1}^{M} h_m(\omega) \qquad (7)$$

This simplification leads to a control variates formulation of RFA as a sum of control variate estimates for each token:

$$\text{RFA}(q_n, K, V) = g(\omega) - \hat{\beta}(\omega) \left(h(\omega) - \mathbb{E}\left[h(\omega)\right]\right) = \sum_{m=1}^{M} g_m(\omega) - \hat{\beta}(\omega) \left(h_m(\omega) - \mathbb{E}\left[h_m(\omega)\right]\right)$$

$$(8)$$

While RFA employs a single $\beta(\omega)$ shared across all tokens, this approach can be generalized to utilize multiple coefficients. In the limit, where each token has its own coefficient, each control variate coefficient can be independently optimized to minimize variances, resulting in an exact recovery of Softmax attention. To balance approximation quality and computational efficiency, Zheng et al. (2023) propose locally shared coefficients.

EVA attention incorporates a local attention component (for $m \in E$), which can be interpreted as locally optimizing the coefficients for individual tokens, as well as local RFA estimates of subsets of tokens ($m \in \mathcal{P}_c$) reweighed by their corresponding true attention score. The local set $E$ and the disjoint subsets $\{\mathcal{P}_c\}_{c=1}^{C}$ collectively comprise the full set of tokens over which attention is applied. Thus, EVA attention is formulated as:

$$\text{EVA}(q_n, K, V) := \tilde{g}(\omega) = \sum_{m \in E} \tilde{g}_m(\omega) + \sum_{c=1}^{C} \sum_{m \in \mathcal{P}_c} \tilde{g}_m(\omega)$$

$$= \sum_{m \in E} \frac{\exp(\boldsymbol{q}_n^T \boldsymbol{k}_m)}{Z} \boldsymbol{v}_m + \sum_{c=1}^{C} \frac{\sum_{m \in \mathcal{P}_c} \exp(\boldsymbol{q}_n^T \boldsymbol{k}_m)}{Z} \frac{\sum_{m \in \mathcal{P}_c} \boldsymbol{g}_m(\omega)}{\sum_{m \in \mathcal{P}_c} \boldsymbol{h}_m(\omega)}$$

$$(9)$$

For practical implementation, the normalization constant requires approximation, as computing it would necessitate summing over all Keys in the context, leading to quadratic complexity in the attention:

$$Z = \sum_{m \in E} \exp(q_n^T k_m) + \sum_{c=1}^{C} \sum_{m \in \mathcal{P}_c} \exp(q_n^T k_m) \approx \sum_{m \in E} \exp(q_n^T k_m) + \sum_{c=1}^{C} \exp(q_n^T \tilde{k}_c) \qquad (10)$$

Furthermore, to efficiently compute $\beta_c(\omega)$, EVA attention employs a single random sample (i.e., $S = 1$), which eliminates the dependency of $\beta_c(\omega)$ on the query $q_n$, allowing for precomputation across all queries. For a comprehensive derivation and detailed explanation of term calculations, we refer the reader to Zheng et al. (2023).

## 2.4 FLASHEVA

We now demonstrate that EVA attention can be reformulated as Softmax attention over a modified set of keys and values, encompassing both the original keys from the local chunk and derived keys and values from the random feature approximation. By substituting the normalization constant $Z$ from Equation 10 into Equation 9, we can express EVA attention as:

$$\text{EVA}(\mathbf{q}_n, \mathbf{K}, \mathbf{V}) \approx \frac{\sum_{m \in E} \exp(\mathbf{q}_n^\top \mathbf{k}_m) \mathbf{v}_m + \sum_{c=1}^{C} \exp(\mathbf{q}_n^\top \tilde{\mathbf{k}}_c) \hat{\beta}_c(\omega)}{\sum_{m \in E} \exp(\mathbf{q}_n^\top \mathbf{k}_m) + \sum_{c=1}^{C} \exp(\mathbf{q}_n^\top \tilde{\mathbf{k}}_c)} = \text{SoftmaxAttn}(\mathbf{q}_n, \tilde{\mathbf{K}}, \tilde{\mathbf{V}})$$

$$(11)$$

where $i$ ranges over both $m \in E$ and $c \in \{1, \ldots, C\}$. We define the augmented key and value sets as:

$$\tilde{\mathbf{K}} = \{\mathbf{k}_m \mid m \in E\} \cup \{\tilde{\mathbf{k}}_c \mid c \in \{1, \ldots, C\}\} \qquad (12)$$

$$\tilde{\mathbf{V}} = \{\mathbf{v}_m \mid m \in E\} \cup \{\hat{\beta}_c(\omega) \mid c \in \{1, \ldots, C\}\} \qquad (13)$$

Note, in the rest of the paper, we refer to the $\tilde{\mathbf{k}}_c$ and $\hat{\beta}_c(\omega)$ as RFA keys and values, or simply random feature-based tokens. This reformulation allows us to leverage existing optimized attention implementations. In the non-causal setting, we can directly utilize the FlashAttention CUDA kernels (Dao et al., 2022) to optimize FlashEVA's performance. The causal setting, however, presents additional challenges. It necessitates attending separately to past keys and random-feature based keys, requiring a custom attention mask not supported by the standard FlashAttention implementation. To address this, we adapt the FlashAttention Triton kernels[2] to accommodate the required attention mask.

This adaptation facilitates a significant enhancement to the original EVA attention mechanism. Specifically, it enables the implementation of a sliding window for local attention, which has the potential to improve model performance by providing an effectively larger context window for the deeper attention layers in the model.

## 3 EXPERIMENTAL SETUP

Our main experiments leverage the open-source Pythia checkpoints (Biderman et al., 2023) for fine-tuning. This choice provides a consistent family of models at various scales, enabling us to evaluate our method's scalability while maintaining architectural and pretraining consistency.

Pythia was pretrained on the Pile (Gao et al., 2020), an 825 GB corpus of English text designed for language model pretraining. To minimize discrepancies between pretraining and fine-tuning setups that could impact the efficiency of adaptation to a different attention mechanism, we utilize the same dataset for model fine-tuning. Due to copyright disputes rendering the original dataset unavailable, we employ the uncopyrighted version accessible on HuggingFace[3].

We fine-tune the models for $50k$ steps with a sequence length of $2048$ and a total batch size of 16, amounting to approximately $1.6B$ tokens. For larger models, we initially warm up the attention layers for $2k$ steps (using a constant learning rate of $3e-4$) before continuing the fine-tuning of all weights for an additional $48k$ steps (using a one-cycle cosine decay schedule). This approach mitigates the risk of large gradients disrupting the pre-learned weights and enhances training stability.

To accelerate training, we employ custom-implemented Triton kernels for FlashEVA, torch compile, and mixed precision. As these optimizations can cause instabilities in larger models during the initial stages of training due to large gradients, we conduct the first $2k$ steps without mixed precision or custom Triton kernels.

We adapt the Pythia model to incorporate gated output projection in the attention layer, similar to Sun et al. (2023) and Chen et al. (2024). To stabilize training, we clip the values and reduce the variance of the random weights in EVA attention. Additionally, we employ RoPE positional encoding (Su et al., 2023) during fine-tuning with FlashEVA, applied to all tokens prior to the random feature projections. Detailed ablation studies on various model and training choices are presented in the Appendix.

For downstream model performance evaluation, we utilize the LM evaluation harness (Gao et al., 2024) to conduct reproducible assessments. Following prior work (Chen et al., 2024; Biderman et al., 2023), we select six tasks: PIQA (Bisk et al., 2019), WinoGrande (Sakaguchi et al., 2019), WSC (Sakaguchi et al., 2019), ARC-C and ARC-E (Clark et al., 2018), and LogiQA (Liu et al., 2020). To further assess the model's retrieval capabilities, we include two additional tasks: FDA (Arora et al., 2023) and SWDE (Arora et al., 2024a).

To assess inference throughput, we consider a fixed prefix/prompt of 2048 tokens, similar to Gu & Dao, with varying numbers of generated tokens ranging from $1024$ to $10240$. To measure maximum throughput, we explore batch sizes up to 32. For the EVA hyperparameters inference performance trade-off analysis, we constrain the setting to a prefix of 2048 tokens and 1024 generated tokens.

---

[2]Available at: `https://github.com/Dao-AILab/flash-attention/blob/main/flash_attn/flash_attn_triton.py`

[3]`https://huggingface.co/datasets/monology/pile-uncopyrighted`

# 4 RESULTS

## 4.1 FINETUNING PERFORMANCE

Table 1: Comparison of downstream task performance for various attention mechanisms across different model sizes. Results show accuracy scores for six general language understanding tasks (left) and two long-context tasks (right). Bold indicates the best performance for each model size, while underlined values represent the second-best. FlashEVA demonstrates competitive performance with Pythia (full attention) on general tasks while outperforming DiJiang and sliding window attention in most cases.

| Model | WinoGrande | PiQA | WSC | ARC-E | ARC-C | LogiQA | Avg. | SWDE | FDA |
|---|---|---|---|---|---|---|---|---|---|
| Sliding-1B | 0.520 | 0.632 | 0.590 | 0.419 | 0.233 | 0.267 | **0.444** | 0.182 | 0.126 |
| Pythia-1B | 0.521 | 0.610 | 0.590 | 0.407 | 0.253 | 0.272 | 0.442 | **0.635** | 0.579 |
| FlashEVA-1B | 0.500 | 0.608 | 0.557 | 0.407 | 0.240 | 0.261 | 0.429 | 0.112 | 0.012 |
| DiJiang-1B | 0.497 | 0.590 | 0.513 | 0.356 | 0.221 | 0.226 | 0.401 | 0.032 | 0.000 |
| Pythia-410M | 0.509 | 0.596 | 0.559 | 0.392 | 0.231 | 0.269 | **0.426** | **0.629** | **0.517** |
| FlashEVA-410M | 0.510 | 0.597 | 0.566 | 0.379 | 0.237 | 0.250 | 0.423 | 0.074 | 0.004 |
| Sliding-410M | 0.512 | 0.596 | 0.544 | 0.370 | 0.239 | 0.272 | 0.422 | 0.072 | 0.000 |
| DiJiang-410M | 0.506 | 0.588 | 0.533 | 0.348 | 0.221 | 0.232 | 0.405 | 0.029 | 0.000 |
| FlashEVA-160M | 0.512 | 0.582 | 0.510 | 0.347 | 0.225 | 0.251 | **0.405** | 0.060 | 0.001 |
| Pythia-160M | 0.493 | 0.570 | 0.532 | 0.331 | 0.210 | 0.281 | 0.403 | **0.152** | **0.247** |
| Sliding-160M | 0.499 | 0.558 | 0.514 | 0.320 | 0.225 | 0.277 | 0.399 | 0.021 | 0.043 |
| DiJiang-160M | 0.488 | 0.568 | 0.523 | 0.326 | 0.226 | 0.234 | 0.394 | 0.010 | 0.000 |
| FlashEVA-70M | 0.506 | 0.567 | 0.522 | 0.328 | 0.222 | 0.250 | **0.399** | 0.011 | 0.000 |
| Pythia-70M | 0.502 | 0.560 | 0.516 | 0.325 | 0.216 | 0.278 | **0.399** | **0.094** | **0.120** |
| DiJiang-70M | 0.506 | 0.558 | 0.512 | 0.325 | 0.217 | 0.238 | 0.393 | 0.001 | 0.000 |
| Sliding-70M | 0.497 | 0.544 | 0.491 | 0.309 | 0.216 | 0.250 | 0.384 | 0.026 | 0.005 |

The experimental results demonstrate that FlashEVA effectively recovers the performance of Pythia on the majority of downstream tasks across various model sizes, ranging from 70M to 1B parameters. Notably, the average performance across six downstream tasks is nearly identical to the Pythia baseline, despite utilizing only 1.6B tokens for finetuning. Furthermore, FlashEVA consistently outperforms DiJiang across all model sizes, suggesting that the random feature approach employed by FlashEVA provides a more effective approximation of Softmax attention.

However, an exception is observed in retrieval-focused tasks, where the pairwise interaction facilitated by full Softmax attention appears to be crucial. In these tasks, while FlashEVA marginally outperforms DiJiang, it exhibits significantly lower performance compared to both the Softmax and sliding window attention baselines. This suboptimal performance can be attributed to the absence of a sliding window mechanism in FlashEVA's architecture. Although we have implemented a FlashEVA variant incorporating sliding window attention (detailed in the Appendix), it was not employed in the current experiments due to training instabilities on larger models, likely stemming from the custom Triton kernel implementation.

Interestingly, the sliding window attention baseline exhibits remarkably robust performance, recovering a substantial portion of Softmax attention's capabilities, with a more pronounced performance gap in retrieval tasks (but higher than FlashEVA or DiJiang). This phenomenon may be partially explained by the limited finetuning of 1.6B tokens, which, while sufficient for adaptation to sliding window attention, may be suboptimal for larger models to fully leverage the FlashEVA or DiJiang attention mechanisms. It is noteworthy that FlashEVA outperforms sliding window attention on smaller models, with this performance differential diminishing as model size increases. Additionally, the sliding window in the attention effectively expands the context window of deeper attention layers, resulting in a more expressive model.

Bick et al. (2024) recently proposed distilling transformers into Mamba architectures. To compare their approach with our method, we fine-tuned the Phi-1.5-1.3B model into FlashEVA and DiJiang

architectures. As evident from the results in Table 2, the distillation approach significantly outperforms our fine-tuning method. However, it is important to note several key differences in the experimental setup. Firstly, their method used twice the amount tokens for training. Secondly, their approach leveraged multiple stages in the process and employed a specialized distillation objective, potentially enabling more effective learning. FlashEVA's improvements are orthogonal to the training setup employed in the training approach, so we hypothesize that combining FlashEVA with a distillation training objective could yield competitive performance.

Table 2: Downstream task performance comparison of Phi-1.5 model variants. Results show accuracy percentages for various benchmarks, as well as their average.

| Model | WinoG. | ARC-E | ARC-C | PiQA | HellaS. | Lamb. | Avg. |
|---|---|---|---|---|---|---|---|
| Phi-1.5-1.3B | 73.4 | 75.6 | 48.0 | 76.6 | 62.6 | 53.4 | 64.9 |
| Phi-Mamba-1.5B | 71.7 | 74.0 | 44.1 | 75.5 | 60.2 | 50.1 | 62.6 |
| Phi-Mamba-1.5B (stage 3) | 62.8 | 64.3 | 27.8 | 75.6 | 52.6 | 43.8 | 54.5 |
| Phi-EVA | 54.7 | 47.8 | 28.9 | 72.5 | 51.1 | 34.3 | 48.2 |
| Phi-DiJiang | 51.2 | 43.1 | 26.3 | 69.2 | 41.8 | 19.3 | 41.8 |

## 4.2 INFERENCE THROUGHPUT AND MEMORY USAGE

Figure 2 presents a comparative analysis of inference throughput and peak GPU memory utilization across various attention mechanisms for different generation lengths. Despite FlashEVA's theoretical design incorporating a cache that scales with sequence length, our empirical observations reveal that peak memory consumption occurs during prefix computation. In practical scenarios, FlashEVA thus exhibits peak memory usage comparable to Sliding Window Attention, while DiJiang Attention demonstrates the lowest memory footprint. Notably, FlashEVA achieves a significant reduction in peak memory usage, up to 5-fold—compared to the vanilla Transformer.

Our results further indicate that FlashEVA surpasses DiJiang Attention in terms of throughput, while marginally underperforming Sliding Window Attention. This performance differential can be attributed to two factors: (1) the reduced number of sequential operations required for token computation in Sliding Window Attention, and (2) the generally memory-bound nature of inference tasks. Nevertheless, FlashEVA demonstrates substantial throughput improvements over the vanilla Transformer, achieving a 2-fold increase when generating 1024 tokens and up to a 6.7-fold improvement for 10240 token generation tasks.

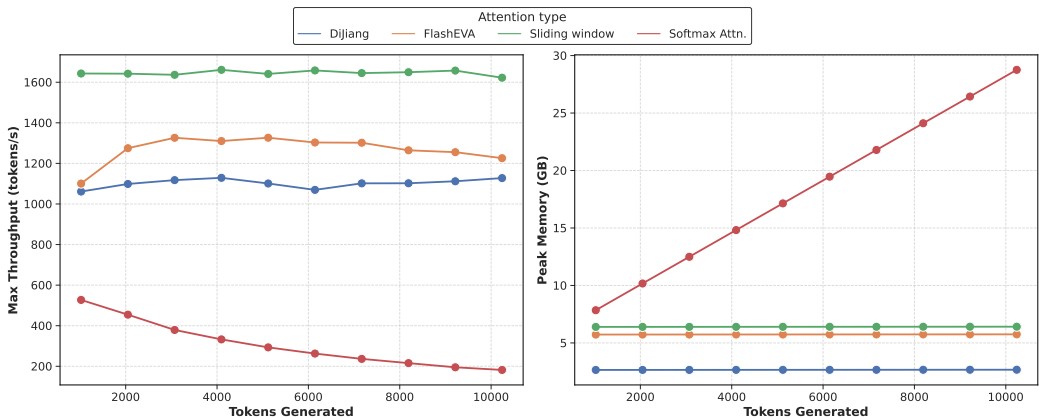

Figure 1: Comparative analysis of maximum throughput and peak GPU memory usage across different attention mechanisms and generation lengths.

### 4.3 FLASHEVA INFERENCE PERFORMANCE TRADEOFF

(Flash)EVA introduces two key hyperparameters that influence its performance and computational requirements: the local attention window size and the number of keys/values compressed into a single random feature-based token. Generally, larger local attention windows or reduced token compression result in increased computational costs.

Our findings demonstrate that FlashEVA can substantially reduce peak memory usage while maintaining competitive performance. Certain configurations decrease peak memory by up to 50% with only a 0.5% reduction in average performance on downstream tasks (excluding retrieval-focused tasks). Moreover, we observe up to a 50% increase in total inference throughput with minimal performance degradation. It is noteworthy that configurations with larger local attention windows exhibit superior performance on retrieval-focused tasks. However, we do not include these tasks in the reported average accuracy, hypothesizing that hybrid models may offer a more favorable trade-off for retrieval task performance.

Interestingly, the configuration used for our main results (local window size 256, RFA chunk size 16) is suboptimal in terms of inference performance. Our analysis suggests that configurations $(256, 8)$ or $(512, 8)$ may offer a more optimal balance between performance and computational efficiency.

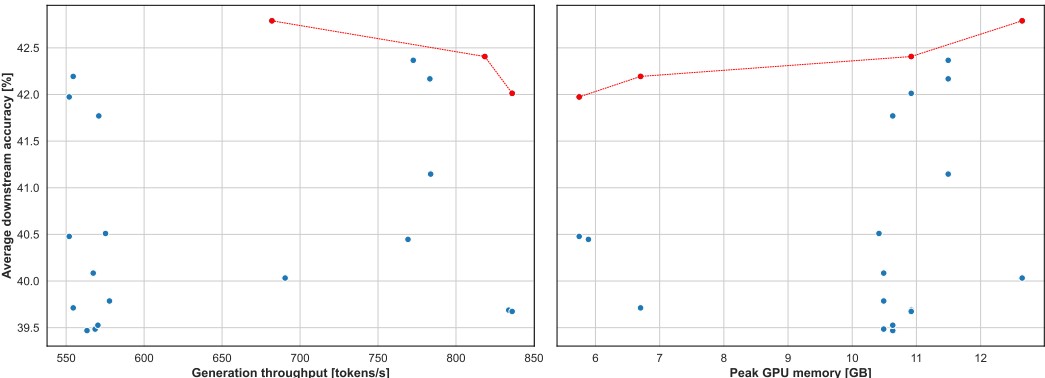

Figure 2: Performance trade-offs between average downstream accuracy and generation throughput or peak GPU memory for various FlashEVA-410M configurations. Each point represents a unique combination of local attention window size and RFA chunk size.

### 4.4 FLASHEVA ATTENTION SPEEDUP

We benchmark the time to run a forward and backward pass of the (Flash)EVA attention layer and compare it to FlashAttention2 kernels. We evaluate at different sequence lengths, adjusting batch sizes to maintain a constant total token count. FlashEVA used a local attention window of size 256 and another 128 random-feature based tokens.

Despite its theoretical promise of improved computational complexity, EVA attention encounters challenges similar to previous linear attention implementations, where practical performance gains fail to materialize, particularly when compared to the efficient FlashAttention implementation. FlashEVA attention does achieve speedups over FlashAttention2 for longer sequences due to its lower computational complexity, up to 2.2x on for the 16k sequence length. However, for shorter sequences, the overhead associated with random feature computation results in slower performance.

## 5 CONCLUSION

This paper presents FlashEVA, an efficient implementation of EVA attention, demonstrating its application for addressing memory constraints in traditional Transformer models during inference. Our experiments reveal that EVA attention enables effective fine-tuning of Transformer models with minimal performance degradation, particularly when retaining a few attention layers.

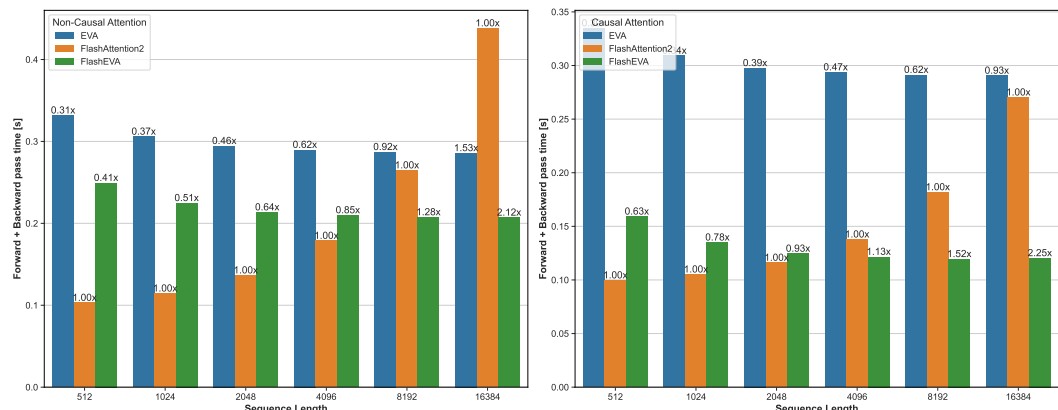

Figure 3: Comparison of forward and backward pass execution times for (Flash)EVA attention layer and FlashAttention2, including both causal and non-causal attention variants. Results are reported for a constant number of chunks $C$ across different sequence lengths, varying chunk size accordingly. Additional results with fixed chunk size and varying $C$ are presented in the Appendix, yielding qualitatively similar outcomes.

Unlike state-space models such as Mamba, EVA attention still requires maintaining a cache that grows with sequence length, however, it is orthogonal to other KV cache compression techniques. As the compression in EVA attention is integral to the forward pass rather than a post-hoc operation, we anticipate high compatibility with existing KV cache compression methods, potentially leading to further reductions in KV cache overhead. This synergistic exploration remains a promising avenue for future work.

Our work on FlashEVA attention represents a significant advancement towards more efficient Transformer models, addressing critical challenges in large language model inference. By balancing performance preservation with memory efficiency, we contribute to ongoing efforts to enhance the accessibility and applicability of advanced language models across diverse computational environments and use cases.

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

# A  RELATED WORK

**Linear transformers** Linear transformers were introduced by (Katharopoulos et al., 2020) where they also showed their formulation as RNNs. Following works Peng et al. (2021); Zheng et al. (2022); Qin et al. (2022); Choromanski et al. (2022), focused on improving the features maps used in linear transformers to close the gap to Vanilla transformer. Subsequent works added also architectural improvements, such as output gating of the attention Qin et al. (2024a); Sun et al. (2023); Hua et al. (2022), and (learned) decay of the hidden state Yang et al. (2023); Ma et al. (2023); Peng et al. (2023). Despite closing the performance gap to transformers on most tasks, there are still fundamental limitations of linear transformers, especially when it comes to retrieval tasks and in context learning (Arora et al., 2024b; Merrill et al., 2024; Akyürek et al., 2024). Finally, (Dao & Gu, 2024) also showed that there exists a connection between linear transformers and specific subset of SSM models.

**State Space Models** Deep State Space Models were introduced as an efficient architecture to model long sequences. S4, DSS showed promising results on long range synthetic tasks. Later models such as GSS (Mehta et al., 2022), BiGS (Wang et al., 2023), and H3 (Fu et al., 2023) introduced gating as a way to increase the expressivity of the model (i.e. increase the interactivity between tokens). The newest generation of models (Mamba (Gu & Dao), GLA Yang et al. (2023), DeltaNet (Yang et al., 2024b), HGRNN (Qin et al., 2024b), xLSTM(Beck et al., 2024)) introduce input dependent SSM parameters, which does not allow the SSM to be expressed as a convolution over the input, however, it can still be computed efficiently with parallel scan. These models are achieving comparable performance to Transformers on language modeling tasks, and crucially, have significantly higher throughput, thanks to the fact, that they do not require to store the full past context on inference, but rather only the hidden state.

**Hybrid Models** Currently, the biggest gap in performance between SSMs/Linear transformer models and Transformer models is the performance on (in context) retrieval tasks (Wen et al., 2024; Jelassi et al., 2024). Consequently, many of the recent models propose to use a combination of attention and recurrent layers in the same model Zancato et al. (2024). Griffin (De et al., 2024) uses a combination of Linear Recurrent Units (Orvieto et al., 2023) with sliding window attention layers, matching the performance of Llama-2, while achieving lower latency and higher throughput on inference in models of size up to 14B parameters. Similarly, Dao & Gu (2024); Waleffe et al. (2024); Glorioso et al. (2024) use a combination of Mamba layers and Attention layers to achieve same or better performance than transformer based baselines.

**Distilling transformers into RNNs** This idea has been initially investigated by Kasai et al. (2021b), where they replaced the Softmax in the attention with learnable feature maps for the queries and keys, composed of a one layer MLP with ReLU activation and then finetuned the pretrained transformer model. They obtained substantial memory savings and inference speedup, however, the performance lagged the pretrained transformer performance on language modelling and machine translation tasks. (Zhang et al., 2024) improved on this by finetuning the transformer into a linearized transformers, where an additional loss terms is used to match the linearized attention to the softmax attention. However, this method required having access to teh full attention matrix of the base transformer during training, which is computationally expensive. Instead, recent works (Bick et al., 2024; Wang et al., 2024) apply more involved distillation methods to distill Transformers into (hybrid) Mamba based models with as few as 3B tokens. They show competitive performance to Transformer on downstream tasks on models up to 7B parameters.

**KV cache compression** The autoregressive nature of language models requires storing the full past context to produce the next new token. To speed up inference, caching key-value states (KV-Cache) in memory is a simple yet effective technique Pope et al. (2022), however, this necessitates a large amount of memory. Consequently, there are several lines of research that looked into compressing KV cache. Quantization based techniques Zhao et al. (2024); Yue et al. (2024); Zhang et al. (2023); Liu et al. (2023b) aim to reduce the memory footprint needed by each token in the cache. Eviction based techniques seek ways to keep only a subset of tokens in the KV cache without sacrificing model performance significantly Jiang et al. (2024; 2023); Han et al. (2024). Additionally, some works looked at compressing the cache via low rank projection of the tokens. Unlike these approaches, our method compresses the cache through summarizing chunks of tokens into single tokens as part of the attention mechanism itself.

## B ABLATIONS

### B.1 EVA LOCAL WINDOW ATTENTION VARIANTS

While the original EVA implementation employs local attention with potentially overlapping windows, recent approaches favor sliding window attention due to its ability to extend the model's effective context window with depth. Our Triton kernel implementations support both variants, prompting a comparative analysis of their performance.

We finetuned the Pythia-70M checkpoint on the Pile dataset for $50k$ steps, utilizing a learning rate of $3e-4$ with a one-cycle cosine decay schedule and $1000$ steps of warmup. The AdamW optimizer was employed with parameters $(0.9, 0.95)$ and a weight decay of $0.1$. The total batch size was set to $16$, with a sequence length of $2048$.

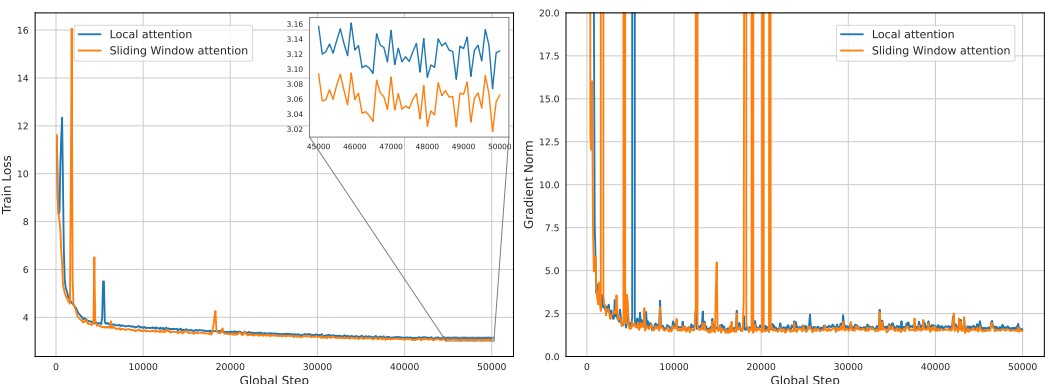

Figure 4: Comparison of training loss and gradient norm during finetuning of CausalEVA with local attention and sliding window attention. The sliding window variant achieves marginally lower loss for the 70M model, but exhibits more unstable gradients with frequent large spikes in norm. The local attention variant demonstrates occasional loss spikes, potentially attributable to the random weight sampling in EVA attention.

### B.2 IMPACT OF RANDOM FEATURE SAMPLING DISTRIBUTION

While Zheng et al. (2023) employ a standard normal distribution centered around the $\mu_c$ vector for each group as the proposal distribution for random features, we observed that this approach leads to instabilities during the training of larger models. To mitigate this issue and stabilize the training process, we introduce a clipping and downscaling mechanism for the distribution width. This modification results in smoother training trajectories with fewer spikes. We define the modified sampling distribution $q_c(\omega)$ as:

$$q_c(\omega) := \lambda \cdot \text{clip}_{[-1,1]} \left( \mathcal{N}(\omega; \mu_c, I) \right) \tag{14}$$

where $\lambda = 0.1$ is the scaling parameter, and $\text{clip}_{[-1,1]}(\cdot)$ denotes the clipping operation that constrains values to the interval $[-1, 1]$. Figure 5 illustrates the effectiveness of our proposed sampling method by comparing the training loss trajectories of models initialized with clipped and unclipped weight distributions.

### B.3 WARMING UP NEW WEIGHTS BEFORE FINETUNING

During the transition from a standard transformer to a linearized variant, we observed that while the MLP is directly transferable, the attention layer undergoes significant changes. This discrepancy can lead to large gradients, potentially pushing weights away from their learned optima and, in larger models, causing numerical instabilities manifesting as NaN values during training. To mitigate these issues, we investigated various strategies for warming up the new weights utilized by FlashEVA or DiJiang attention mechanisms.

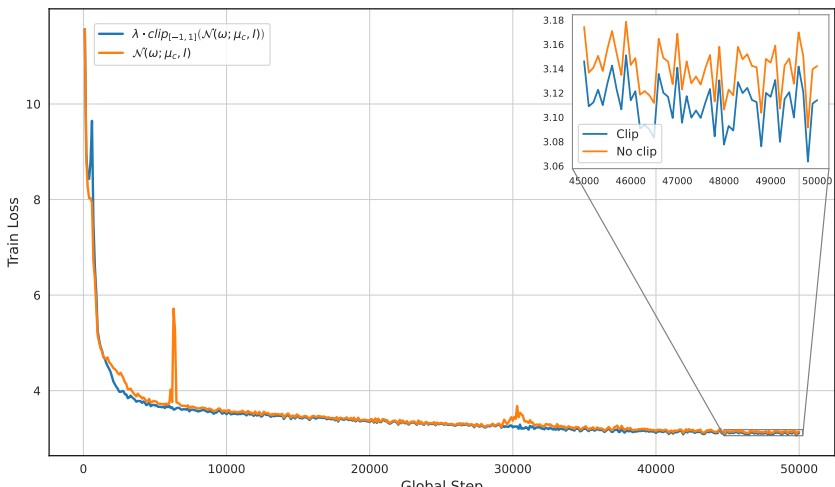

Figure 5: Comparison of training loss trajectories for models initialized with clipped and unclipped weight distributions, demonstrating the stabilizing effect of our proposed sampling method.

We conducted experiments using the 70M Pythia model with CausalEVA attention, employing random weight clipping as defined in Equation equation 14. Our warmup protocol consisted of 2000 steps with a 'warmup and constant' learning rate schedule, followed by $48k$ steps using the configuration from our main experiments. Similar outcomes were observed for DiJiang finetuning, hence we omit those results for brevity. Based on these findings, we adopted the practice of warming up attention layer weights with full precision for experiments involving larger model sizes. Figure 6

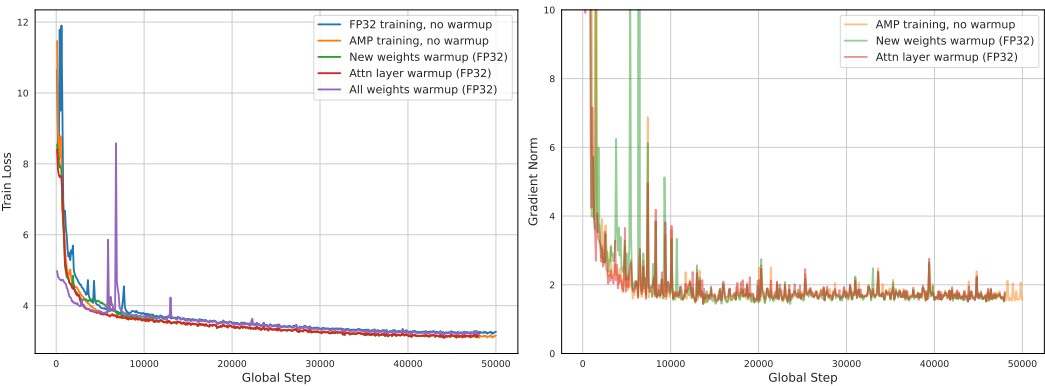

Figure 6: Comparison of training loss stability and gradient norms across different training strategies. Warming up attention layers prior to finetuning yields the most stable training and minimizes gradient norm spikes, informing our approach for larger-scale model experiments.

illustrates the comparative stability of training loss across various training configurations. We also present gradient norm trajectories for three key options, as their training curves exhibit notable similarities. The results demonstrate that warming up attention layers before proceeding with finetuning leads to the most stable training process and minimizes gradient norm fluctuations. Consequently, we adopted this approach for our experiments with larger model architectures.

## B.4 FINETUNING DURATION

Recent work by Bick et al. (2024) demonstrated effective performance when distilling a transformer into a Mamba model using 3B tokens for finetuning. Motivated by this, we investigate the impact of finetuning duration on FlashEVA's performance. We finetune the Pythia 70M model

using consistent training setups and hyperparameters across experiments, varying the total steps: $[50k, 100k, 200k, 350k, 500k]$. We evaluate the minimum training loss achieved and the average accuracy on the downstream tasks (excluding retrieval focused tasks).

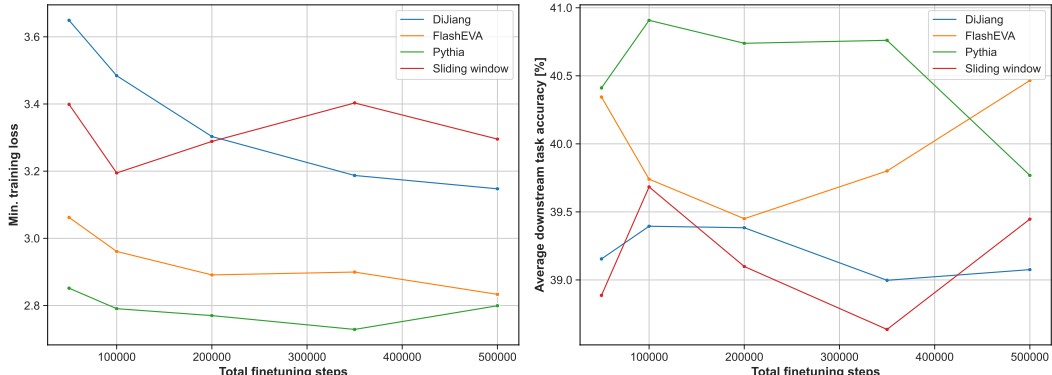

Figure 7: Comparison of minimal training loss and average downstream task performance across varying finetuning durations. While extended finetuning leads to continued decrease in training loss, downstream performance remains relatively stable, suggesting limited benefits from prolonged training.

Our results, illustrated in Figure 7, reveal that downstream performance remains relatively consistent across all tested finetuning durations, with variations falling within the range of random seed fluctuations. Consequently, we conduct our main experiments using 50k steps, corresponding to approximately 1.6B tokens.

## C  ADDITIONAL RESULTS

### C.1  EVA KERNEL SPEEDUP

For the attention layer speed comparison, we consider sequence lengths $[512, 1024, 2048, 4096, 8192, 16384]$, and we vary the batch size $B = \frac{16384 B_{min}}{L}$, where $L$ is the sequence length and $B_{min}$ is the minimum batch size at the longest sequence length. We considered $B_{min} \in [1, 2, 4, 8]$, and report the results for $B_{min} = 8$ (however, the results are qualitatively the same, only the exact speedup values differ slightly)

We report here also the results for the attention layer speedup for the setting where the size of the chunk that is used to compute the local control variates is kept fixed as the sequence length is varied.

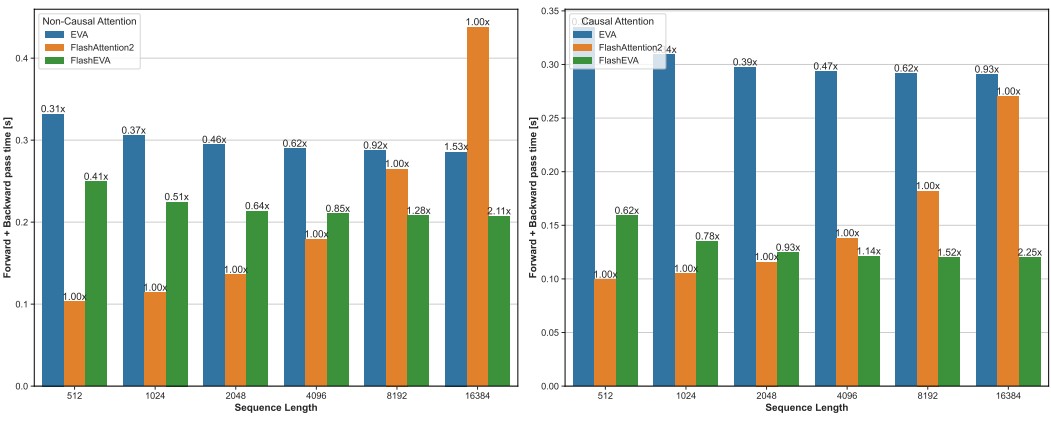

Figure 8: Time to run the forward and backward pass of the (Flash)EVA attention layer compared to FlashAttention2 for the setting where the chunk size is kept constant.