# OpenReview forum: "FlashEVA: Accelerating LLM Inference via Efficient Attention"
_ICLR.cc/2025/Conference — Submitted to ICLR 2025_

### Official Review · Reviewer_x8Hx · 2024-10-29

**Soundness:** 3
**Presentation:** 3
**Contribution:** 1
**Rating:** 3
**Confidence:** 5

**Summary:**

The paper present FlashEVA, an efficient implementation of EVA and demonstrate how to finetune transformers to adapt to FlashEVA attention. The experimental results show that it can accelerate LLMs up to 6.7x with little performance drop.

**Strengths:**

- The paper is well-written and easy to understand.

- The experimental results are rich.

**Weaknesses:**

- I donot see any novelty in this paper. Eq.11 is derived from Eq.9 and Eq.10 in the background section and it seems that the author only define the augmented key and value set.

- Overall, this paper does not provide anything new, and it is more like an experimental report rather than an academic paper.

I think this paper is not ready for submitting to ICLR.

**Questions:**

See weaknesses above.

---

### Official Review · Reviewer_vRj4 · 2024-11-03

**Soundness:** 3
**Presentation:** 2
**Contribution:** 2
**Rating:** 5
**Confidence:** 4

**Summary:**

- This paper proposes an efficient attention implementation via control variants method using custom CUDA and Triton kernels
- can be finetuned from transformer models with only 1.5b tokens
- show higher inference throughput and lower memory.
- suffers in retrieval focused tasks

**Strengths:**

- The motivation, and the background about different forms of attentions are clear.
- Extensive experiment results including different down-stream tasks.
- The proposed methods achieve obvious better throughputs and memory consumption comparison compared with EVA and flash attention.
- The proposed method can be finetuned from a standard attention models which makes it easier to use.

**Weaknesses:**

- The contribution of this work is an incremental work based on EVA. It is not a new algorithm but an efficient implementation of EVA.
- Although the background of RFA, EVA are clearly explained, some background about flash attention could be included since it is more related and if the reader is not familiar.
- In addition, more details should be given about the CUDA implementation, such as pseudo code and how the custom attention mask is achieved. Current presentation about the flashEVA is too simple.

**Questions:**

This method is claimed to suffer in retrieval focused tasks. I wonder what if the reason, is this due to the nature of linear attention?

---

### Official Review · Reviewer_H6Vo · 2024-11-03

**Soundness:** 2
**Presentation:** 2
**Contribution:** 2
**Rating:** 3
**Confidence:** 3

**Summary:**

The paper introduces FlashEVA, an efficient implementation of EVA to improve Transformer model performance during inference by reducing memory usage. FlashEVA allows fine-tuning of Transformers with few tokens. While it excels in many applications, it has limitations in retrieval-focused tasks. The method also offers adjustable settings for balancing throughput and accuracy, enhancing flexibility for different use cases.

**Strengths:**

This paper is well-organized and clear, effectively presenting complex ideas in efficient attention mechanisms. Background and motivation are well-integrated, and the experimental results are systematically laid out, with tables and figures that clarify performance gains. The discussion of trade-offs and limitations shows a balanced approach, enhancing the paper’s readability and impact.

**Weaknesses:**

A primary limitation of this paper is its lack of significant novelty beyond the existing EVA framework. While FlashEVA offers efficiency gains, these improvements are largely a result of optimizing existing CUDA/Triton kernels rather than introducing new concepts. As such, the contribution may appear incremental, particularly given the relatively modest improvements in throughput and memory efficiency.

While the paper briefly compares FlashEVA with the Mamba model, it does not thoroughly examine their differences or provide a clear rationale for preferring FlashEVA. Mamba significantly outperforms FlashEVA across most tasks, even with fewer computational resources. To provide a more balanced view, the authors could explore the advantages FlashEVA may offer over Mamba, or discuss potential benefits of integrating aspects of both methods. Such a discussion could help clarify FlashEVA’s unique contribution and when it might be favored over Mamba, thereby enhancing the overall value of the work.

FlashEVA demonstrates impressive gains with long-sequence generation but shows limited improvement for shorter sequences. This restriction reduces the model's scalability, especially in applications requiring shorter or mixed-length sequences. The authors could mitigate this issue by optimizing the random feature computation or implementing adaptive techniques that reduce computational overhead for shorter sequences.

**Questions:**

Please address the questions raised in the weaknesses section.

---

### Official Review · Reviewer_Y2bY · 2024-11-04

**Soundness:** 2
**Presentation:** 3
**Contribution:** 2
**Rating:** 3
**Confidence:** 5

**Summary:**

This paper proposes FlashEVA, an efficient implementation of EVA, aimed at addressing the high memory demands of Transformer models  during inference. FlashEVA allows for fine-tuning of large language models (LLMs) with minimal performance degradation while significantly reducing memory usage and increasing throughput. The method achieves this by maintaining a compressed cache of past context, which is less memory-intensive than computing the prefix during inference. However, it shows limitations in retrieval-focused tasks. The implementation offers adjustable hyperparameters to balance throughput, accuracy, and memory usage, providing flexibility for various applications.

**Strengths:**

+ The paper is well organized and presented.
+ Experimental results that the adjustable hyperparameters allow for a trade-off between throughput, accuracy and memory usage.
+ FlashEVA is compatible with existing optimized attention implementations and can leverage CUDA kernels for performance optimization.

**Weaknesses:**

- The proposed method is overly simplistic and unimpressive. It looks like an implementation of FlashAttention with EVA which is already proposed.
- The experimental results are not persuasive since it doesn’t show the advantages compared to Dijiang and Sliding window as in Figure1. Instead, it is only a trade-off between Dijiang and Sling window.

**Questions:**

- How about the performance when the model is larger such as 7B, 13B or 70B?
- Can the method compared to FlashAttention3 or combined with it?

---

### Author Response · Authors · 2024-11-26
**Addressing novelty criticism**

We would like to thank all the reviewers for their constructive feedback. We are happy to hear the that you have found the paper easy to read, and the experimental results comprehensive. On the other hand, most have raised objections regarding the novelty of the method, which is what we would like to address.

We agree with the reviewers that the proposed method is simple and can appear incremental over the existing EVA approach. Nevertheless, we would argue that this does not mean that it is not useful and of interest to the community.

First, we propose to apply the EVA framework in a different setting as the original authors. While they have mainly focused on the training from scratch, we focus on adaptation of pretrained transformers into linearized variants in order to improve model inference efficiency. For this, we leverage the EVA framework due to its Softmax attention approximation guarantees.

Futhermore, we would like to highlight two crucial problems that have limited an adoption of efficient attention approaches, especially for the model inference.

1. There is a need for efficient Implementations in order to be able to materialize the theoretically promised speedups of these methods. By showing that EVA attention can be reformulated as a Softmax attention over modified set of keys and values, we can leverage community improvements to obtain a very efficient implementation of the EVA attention. This allows FlashEVA to optimize inference in two ways: for the prompt processing, it leverages the lower computational complexity of the efficient attention, while for the autoregressive generation/decoding, it benefits from the lower memory footprint, which is the main bottleneck in the decoding process.

2. Linearized attention approaches have generally faced a performance degradation compared the transformers using standard softmax attention. EVA attention has solid theoretical grounding and with its hyperparameters we can tune how well we will approximate softmax attention. In fact, our method consistently outperforms Diljang attention over all model scales, showing that the EVA framework is adequate for this task. Nevertheless, as mentioned, all linear attention approaches will suffer a decreased performance on the retrieval focused tasks, as that is a fundamental limitation of the linearized approach [1]

Given the above points, we feel that the FlashEVA method is a contribution of interest to the wider community.

Furthermore, we would like to expand and contextualize further the main experimental results. The main comparison baseline is the Diliang attention, a recent approach showing comparable performance to Softmas attention transformers on inference, with substantial efficiency improvements. As can be seen, our method outperforms consistently the Dijiang attention for all model scales. In fact, at larger model scales, in some tasks we achieve up to 5% better accuracy. Indeed, one of the limitations of the Dillang method we observed was its variability in performance, drastically underperforming the baseline Transformer in certain tasks.

We additionally compared FlashEVA to a Sliding Window attention baseline, with a larger sliding window (to match the additional Keys and Values in the FlashEVA approach). Note, in the main results of FlashEVA we did not use a sliding window approach, which would likely improve performance (the reason was that we observed some numerical instabilities when testing this on larger models, so for the sake of saving resources, we only used the local window approach in the main experiments). Only in the largest 18 model size we observe some performance regression compared to that baseline, however, that is mainly driven by subpar performance on the WSC and Winogrande tasks, which generally contain quite short sequences. For those sequences, the Sliding Window attention basically acts like full attention over the whole sequence (also seen in the fact that it basically achieves the same performance as the full attention variant)

Finally, in terms of downstream task performance, we also do not expect to achieve higher performance as the baseline transformer, as we are merely trying to efficiently approximate the full attention.

We have thus shown a method that matches better baseline Transformer on downstream tasks, which has a principled way of trading off some performance for efficiency, and has practical speedups in the model generation.

Having said all that, we understand that the novelty criticism is valid, and ICLR might thus not be the best venue for this submission. We will take into account all the useful feedback (e.g. improving the paper writing to describe the FlashEVA better, introduce FlashAttention, more concrete comparison to Mamba, etc.) in any future submission.

[3] Jelassi, S., Brandfonbrener, D., Kakade, S. M. & Malach, E. Repeat After Me: Transformers are Better than State Space Models at Copying. (2024) ArXiv: http://arxiv.org/abs/2402.01032

---

### Meta-Review · Area_Chair_JVrT · 2024-12-12

**Metareview:**

The paper "FlashEVA: Accelerating LLM Inference via Efficient Attention" introduces FlashEVA, an efficient implementation of Efficient Attention via Control Variates (EVA) to reduce the memory and computational demands of Transformer models during inference. FlashEVA achieves up to 6.7x throughput improvement and 5x memory reduction, enabling fine-tuning with minimal data while maintaining strong performance across general NLP tasks. It provides flexibility through hyperparameter tuning, balancing speed, accuracy, and memory usage, and demonstrates compatibility with existing optimized attention mechanisms like FlashAttention. FlashEVA underperforms on retrieval-focused tasks due to the lack of a sliding window mechanism and faces training instabilities in larger models.  The paper is well organized and written. However, the technical novelty of this paper is still limited since it looks like a modification of an existing method. While the paper briefly compares FlashEVA with the Mamba model, it does not thoroughly examine their differences or provide a clear rationale for preferring FlashEVA. I think the paper could be further improved according to the suggestions for the future submission.

**Additional Comments On Reviewer Discussion:**

The authors do not reply to each reviewer independently, and therefore no discussions are conducted.

---

### Decision · Program_Chairs · 2025-01-22

Reject